# Computational Profiling of Monoterpenoid Phytochemicals: Insights for Medicinal Chemistry and Drug Design Strategies

**DOI:** 10.3390/ijms26167671

**Published:** 2025-08-08

**Authors:** André Nogueira Cardeal dos Santos, Paulo Elesson Guimarães de Oliveira, José Ednésio da Cruz Freire, Sara Araújo dos Santos, José Eduardo Ribeiro Honório Júnior, Claudia Roberta de Andrade, Bruno Lopes de Sousa, Wildson Max Barbosa da Silva, Ariclécio Cunha de Oliveira, Vânia Marilande Ceccatto, José Henrique Leal Cardoso, Adélia Justina Aguiar Aquino, Andrelina Noronha Coelho de Sousa

**Affiliations:** 1Experimental Physiology Laboratory, Superior Institute of Biomedical Sciences, State University of Ceará, Fortaleza 60714-903, CE, Brazil; andrelina.noronha@uece.br; 2Biochemistry and Gene Expression Laboratory, Superior Institute of Biomedical Sciences, State University of Ceará, Fortaleza 60714-903, CE, Brazil; paulo.elesson@aluno.uece.br (P.E.G.d.O.); jednesio@gmail.com (J.E.d.C.F.); vania.ceccatto@uece.br (V.M.C.); 3Biochemistry and Signal Transduction Laboratory, Superior Institute of Biomedical Sciences, State University of Ceará, Fortaleza 60714-903, CE, Brazil; sara.santos@aluno.uece.br (S.A.d.S.); brunolopes.sousa@uece.br (B.L.d.S.); 4Neuroscience and Translational Medicine Laboratory, Christus University Center—Unichristus, Fortaleza 60190-180, CE, Brazil; eduribiologo@yahoo.com.br (J.E.R.H.J.); claudiarandrade@gmail.com (C.R.d.A.); 5Natural Products Chemistry Laboratory, State University of Ceará, Fortaleza 60714-903, CE, Brazil; wildson_max@uvanet.br; 6Endocrine and Metabolism Physiology Laboratory, Superior Institute of Biomedical Sciences, State University of Ceará, Fortaleza 60714-903, CE, Brazil; ariclecio.oliveira@uece.br; 7Electrophysiology Laboratory, Superior Institute of Biomedical Sciences, State University of Ceará, Fortaleza 60714-903, CE, Brazil; lealcard@gmail.com; 8Department of Mechanical Engineering, Texas Tech University, Lubbock, TX 79409, USA; adelia.aquino@ttu.edu

**Keywords:** monoterpenoids, ADME and toxicological prediction, drug-likeness, structure–activity relationship, target prediction

## Abstract

Monoterpenoids are a structurally diverse class of natural products with a long-standing history of therapeutic use. Despite their promising bioactivities, their clinical development has been limited by dose-dependent toxicities, poor pharmacokinetics, and suboptimal drug-like properties. In this work, a comprehensive in silico pipeline was employed to evaluate 1175 monoterpenoid compounds retrieved from ChEBI, aiming to identify structurally diverse candidates that possess favorable drug-like characteristics. A total of 54 molecular parameters were calculated using thirteen computational tools, covering physicochemical parameters, ADMET profiles, and toxicological risk assessments. Stepwise filtering was employed to retain only compounds meeting stringent thresholds across multiple domains, followed by chemoinformatic analysis. Structure–activity relationship mapping and target prediction were subsequently conducted to explore mechanistic plausibility. This workflow led to the identification of seven top-performing monoterpenoids that exhibited ideal physicochemical profiles, high gastrointestinal absorption, low predicted toxicity, and full compliance with medicinal chemistry rules. Notably, target prediction revealed a convergence on GPCRs, enzymatic and nuclear receptors, highlighting potential anti-inflammatory and neuromodulatory effects. The identification of conserved pharmacophores across selected scaffolds further reinforces their translational potential. Our results highlight the value of multi-parameter computational triage in natural product drug discovery and reveal a subset of overlooked monoterpenoids with promising preclinical applications.

## 1. Introduction

Monoterpenoids are a diverse class of natural compounds whose biosynthesis is derived from two isoprene units and are widely represented across plant species [1]. For millennia, these phytochemicals have played a central role in traditional medicinal systems across civilizations, from the Egyptians, Greeks, ancient Chinese, and Indigenous people, being employed for their antiseptic, anti-inflammatory, antioxidant, and calming properties [2]. Classic examples such as camphor were historically used to combat infectious diseases like the Black Death [3], while others such as cineole (eucalyptol), citral, and eucalyptol have been applied in respiratory therapies and food preservation [4].

Although empirical use of monoterpenoids has been partially validated by modern pharmacological studies, their clinical application remains limited due to several critical challenges [3,4,5,6]. Many well-characterized monoterpenoids have been associated with dose-dependent adverse effects that complicate their therapeutic application. Camphor, for example, is a potent neurostimulant capable of crossing the blood–brain barrier and inducing seizures, particularly in pediatric populations, with neurotoxicity reported at doses as low as 50 mg/kg [7]. 1,8-cineole (eucalyptol), commonly employed in respiratory formulations, has been shown to reduce gastric compliance in preclinical models, suggestive of gastrointestinal irritation when administered at higher concentrations [8].

Citral, a major constituent of citrus oils, demonstrates phototoxic potential, with ultraviolet light exposure triggering reactive oxygen species formation and genotoxic damage in skin cells [9]. In addition, *L*-carvone and structurally related terpenoids are classified as dermal sensitizers and have been implicated in allergic contact dermatitis upon repeated topical exposure [10]. These toxicological risks, when coupled with poor aqueous solubility, susceptibility to rapid metabolic degradation [11,12], and variable oral bioavailability [13,14,15], have collectively constrained the advancement of monoterpenoids as drug candidates. As a result, despite their remarkable biological activities and structural diversity, these compounds have often been deprioritized in modern pharmacological pipelines, not due to a lack of potential but rather due to challenges that demand more refined and targeted screening approaches [16].

Computational-aided drug design (CADD) encompasses a range of in silico strategies, including quantitative structure–activity relationships (QSAR) [17], quantitative structure–property relationships (QSPR) [18,19], quantitative structure–biological relationships (QSBR) [20], molecular docking, molecular dynamics simulations [21], pharmacophore modeling [22], and virtual screening [23], that collectively enable a rational, target-oriented evaluation of chemical libraries. These methodologies facilitate the prediction of molecular interactions, bioactivity, and conformational stability, accelerating the identification and optimization of bioactive compounds with favorable pharmacological profiles. The integration of computational methods into early-phase drug discovery has revived interest in these compounds [2,24,25,26]. In silico modeling advances now allow for rapid and systematic evaluation of large chemical libraries, simulating how small molecules are absorbed, metabolized, distributed, and eliminated by the body [27,28,29]. These tools assess pharmacokinetic feasibility, potential toxicity, and chemical responsibility, providing critical foresight into drug-likeness and safety profiles long before in vitro validation [28]. This paradigm shift has positioned computational triage as a central strategy to overcome longstanding barriers in natural product development.

By combining physicochemical analysis, absorption, distribution, metabolism, excretion, and toxicity (ADMET) predictions and medicinal chemistry filters, it is possible to refine molecular libraries based on established pharmacological rules [30]. These criteria guide the identification of compounds that align with physiological constraints, such as optimal membrane permeability, metabolic stability, and target accessibility, while minimizing off-target risks. Collectively, such approaches enhance the likelihood of identifying natural products with clinically relevant profiles [31].

The aim of this study was to identify structurally diverse molecules with ideal drug-like properties through a comprehensive computational screening pipeline to a curated library of 1175 monoterpenoid compounds. Using rigorous multi-parameter filtering, we sought to uncover underexplored candidates that combine favorable physicochemical attributes, pharmacokinetic behavior, and low predicted toxicity. This work highlights the untapped therapeutic potential of lesser-known monoterpenoids and underscores the power of rational, prediction-driven selection strategies in revitalizing neglected natural product scaffolds for modern drug discovery.

## 2. Results

The search performed using the ChEBI Ontology yielded a total of 1175 monoterpenoid compounds (Figure 1, Appendix A). At this initial stage, no specific or well-known monoterpenoid structures were prioritized; instead, the default dataset provided by the PubChem database was comprehensively considered. Of these, 25 compounds were excluded for not meeting the MW criterion, defined as being within the range of 100 to 600 Daltons, a threshold recognized for optimizing membrane permeability and oral bioavailability. Consequently, 1150 molecules advanced to the next stage of analysis. Following this, the presence of nHA and nHD donors was used as the basis for the filter. Molecules exceeding the recommended physiological limits of nHA ≤ 12 and nHD ≤ 7 (a total of 14 molecules) were excluded, reducing the dataset to 1136 molecules. The TPSA was then evaluated, applying the threshold of ≤140 Å^2^, above which passive membrane permeability typically declines. This resulted in the further exclusion of 43 molecules, leaving 1093 compounds. Next, lipophilicity was assessed via the log P parameter. Molecules with log P values exceeding 3 were excluded. This step resulted in a significant reduction of 358 molecules, leaving 735 molecules remaining. Finally, log D at physiological pH (7.4) was applied as an additional refinement, leading to the further exclusion of 27 molecules. This culminated in a subset of 708 compounds selected for subsequent pharmacokinetic evaluations (Figure 1A).

### 2.1. Absorption

Using the Human Intestinal Absorption (HIA) parameter, the initial set of 708 molecules was reduced to 666 after excluding 42 compounds with unfavorable absorption profiles. Subsequently, the permeability of the remaining molecules to Caco-2 cells was evaluated, leading to the exclusion of an additional 71 compounds due to suboptimal permeability, resulting in a dataset of 595 molecules. Following the analysis of interactions with P-glycoprotein (P-gp), the subset was further restricted to 520 compounds, as 75 were identified as P-gp substrates. Filters for P-gp inhibitory compounds (classes I and II) were then applied, resulting in the exclusion of 22 more molecules. Thus, a total of 498 compounds with favorable absorption profiles proceeded to the next evaluation stages (Figure 1B).

### 2.2. Distribution

For the distribution parameters, the application of the plasma protein binding (PPB) filter excluded 60 molecules, reducing the number of candidates from 498 to 438. The subsequent evaluation of the volume of distribution at steady state (VDss) resulted in the exclusion of 37 molecules, leaving 401 compounds selected for further analysis (Figure 1C).

### 2.3. Metabolism

After applying a filter capable of evaluating the inhibitory effects on members of the cytochrome P_450_ enzyme superfamily (CYP_1A2_, CYP_2C19_, CYP_2C9_, CYP_2D6_, and CYP_3A4_), the dataset of 401 candidate molecules was reduced to 365. Among the 36 molecules excluded during this analysis, predictions indicated that 21 were capable of inhibiting CYP_2C19_; 4 were capable of inhibiting CYP_2C9_, CYP_2D6_, and CYP_3A4_; and 11 were predicted to inhibit either CYP_2D6_ or CYP_3A4_ (Figure 2A–C).

### 2.4. Excretion

The excretion profile was assessed by evaluating total systemic clearance, plasma half-life (t½), and potential interaction with the renal Organic Cation Transporter 2 (OCT_2_). Application of these criteria resulted in the exclusion of 6 compounds, reducing the dataset from 365 to 359 molecules selected for subsequent toxicological assessments.

### 2.5. Toxicity

For the toxicological assessment, the Ames mutagenicity test was applied using two approaches: a classification-based filter, which reduced the number of molecules from 359 to 343, and a percentage-based analysis, which led to the exclusion of an additional 21 compounds, resulting in a total of 322 molecules. The subsequent screening for carcinogenic potential caused a substantial reduction, narrowing the dataset from 322 to 130 molecules. Additionally, hepatotoxicity analysis using both qualitative and percentage-based approaches resulted in the exclusion of one compound in the first approach, while the second approach led to the removal of 33 more compounds, yielding a group of 129 molecules. Prediction of the MTD in humans resulted in the exclusion of 15 additional compounds, reducing the dataset to 114 molecules. While cardiotoxicity, evaluated through the potential inhibition of hERG I and II channels, was assessed, all potentially cardiotoxic compounds had already been removed during earlier filtering steps. Finally, the respiratory toxicity assessment excluded 37 additional molecules, leading to a final subset of 44 compounds selected at the conclusion of the toxicological screening (Figure 3).

### 2.6. Medicinal Chemistry Assessment

During the medicinal chemistry evaluation, additional filters were applied as inclusion and exclusion criteria. The Bristol–Myers Squibb (BMS) (New York, NY, USA) filter reduced the number of molecules from 44 to 41. Compounds with alerts for chelating activity and pan-assay interference compounds (PAINS) had already been eliminated in previous phases, requiring no further exclusions at this stage. The remaining molecules (a total of 41 molecules) conformed to acceptable thresholds for quantitative estimate of drug-likeness (QED) and synthetic accessibility or synthetic feasibility (Synth). As the physicochemical criteria were established based on Lipinski’s Rule of Five, all remaining molecules complied with these parameters. However, application of the more restrictive Pfizer filter excluded an additional 13 compounds, reducing the dataset from 41 to 28 molecules. Finally, the Golden Triangle criterion, which integrates key physicochemical properties associated with successful clinical candidates, led to the exclusion of 21 compounds, resulting in a final selection of 7 monoterpenoid molecules with optimal profiles for further consideration as potential drug candidates (Figure 1D).

### 2.7. Top Monoterpenoid Molecules and SAR Analysis

A total of seven compounds, out of the initial set of 1175 screened monoterpenoids, successfully passed all applied filters for physicochemical, pharmacokinetic, toxicological, and medicinal chemistry properties (Figure 4A–G). All compounds were evaluated in their stereoisomeric forms (cis/trans and R/S), and the most stable or bioactive conformer was selected for analysis. Compounds I (15-hydroxyculmorin, CID: 98642738) and II (12-hydroxyculmorin, CID: 139587999) shared a conserved polycyclic core, differing only at substituent positions R1 and R2, as revealed by structure–activity relationship (SAR) analysis (Figure 4H). Compounds III (Xylariterpenoid I, CID: 156582074) and IV (Xylariterpenoid H, CID: 139205363) also presented close structural similarity, with variations mainly in hydroxyl group positioning. Compounds V (Menthoxypropanediol, CID: 5362595) and VI (Trichobasabolin F, CID: 146683370) displayed flexible monocyclic or open-chain systems with polar functional groups, while compound VII (Vulgarole, CID: 101416067) exhibited a distinct fused-ring scaffold. The diversity in carbocyclic architectures, ranging from bridged and fused systems to unsaturated monocyclic rings, is summarized in the ring system classification (Figure 4I), reflecting the conformational and topological variation among the final candidates.

### 2.8. Physicochemical Properties, Toxicological Parameters, and Medicinal Chemistry Compliance of the Seven Final Compounds

All seven selected compounds complied with the full set of physicochemical, toxicological, and medicinal chemistry criteria established in this study. All seven selected compounds complied with the full set of physicochemical, toxicological, and medicinal chemistry criteria established in this study. Their molecular weights ranged from 224 to 230 Daltons. The nHA varied between 2 and 3, while the nHD ranged from 1 to 3. The TPSA values ranged from 37.3 to 49.7 Å^2^, and log P values were between 1.75 and 2.93 (Figure 5A).

In the toxicological assessment, all seven compounds showed predicted probabilities below 0.283 for mutagenicity, carcinogenicity, hepatotoxicity, and respiratory toxicity, and none of the compounds were predicted to inhibit hERG I or II potassium channels (Figure 5B). In addition, all molecules complied with the medicinal chemistry filters of Lipinski, Pfizer, GSK, and Golden Triangle. Their QED scores were greater than 0.66, indicating alignment with favorable drug-likeness profiles (Figure 5C).

### 2.9. Computational Target Profiling and Shared Bioactivity Landscape

Following the submission of the seven final compounds to the TargetNet platform (http://targetnet.scbdd.com/, accessed on 18 June 2025), the predicted protein targets were analyzed individually for each molecule (Appendix A). Applying a probability cutoff of >70%, the primary classes of predicted targets were identified as follows: enzymes (43.42%), G protein-coupled receptors (GPCRs, 18.42%), and nuclear receptors (9.21%), as shown in Figure 6A. An intersection analysis revealed two molecular targets with high interaction probabilities across all seven compounds (Figure 6B). These targets were nitric oxide synthase, inducible (iNOS), and the DNA dC→dU editing enzyme APOBEC-3A. In addition, a broader cutoff was applied to identify targets shared by 5, 6, or all 7 compounds, the results of which are illustrated in Figure 7.

## 3. Discussion

This study employed a comprehensive, multi-parameter in silico strategy to systematically evaluate a large and chemically diverse set of 1175 monoterpenoid compounds, with the aim of identifying promising candidates for further drug development, replacement, or use as drugs. Through successive and rigorously defined filtering stages encompassing physicochemical properties, pharmacokinetics, toxicological profiles, and medicinal chemistry rules, the dataset was refined to a final selection of seven top-ranked molecules. These compounds not only met all the established cutoffs across more than 50 computational descriptors but also exhibited structural diversity and favorable drug-likeness profiles, as evidenced by their QED scores, physicochemical ranges, and compliance with industry-standard medicinal chemistry filters, such as Lipinski, Pfizer, GSK, and the Golden Triangle criteria. Notably, these seven monoterpenoids demonstrated low predicted toxicological risk, including the absence of mutagenicity, carcinogenicity, hepatotoxicity, and cardiotoxicity signals. The SAR analysis revealed that certain candidates shared conserved scaffolds and functional group arrangements, suggesting potential pharmacophores relevant to their predicted bioactivity. Furthermore, computational target profiling identified enzymes, GPCRs, and nuclear receptors as the predominant classes of predicted molecular targets.

The pharmacokinetic behavior and biological response of a drug candidate are governed by a multitude of physicochemical and biochemical variables, making early-stage profiling a critical step in modern drug discovery [32]. Among the first barriers to effective bioactivity are intrinsic molecular features, particularly size and molecular weight, which directly influence passive diffusion across biological membranes and access to intracellular or membrane-bound protein targets. Parameters such as log P, nHA, nHD, and TPSA are widely recognized as central determinants of a compound’s oral absorption, systemic distribution, and excretory profile [33]. In the context of orally administered compounds, first-pass metabolism mediated by cytochrome P_450_ enzymes introduces an additional layer of complexity, as metabolic transformation can generate both active metabolites and potentially toxic byproducts, depending on the functional groups present [34]. Furthermore, the ability of a molecule to induce mutagenicity, carcinogenicity, hepatotoxicity, or cardiotoxicity represents a major bottleneck in drug development, contributing to the high attrition rates observed during preclinical and clinical transitions [35]. In this regard, computational pre-screening platforms based on predictive bioactivity and toxicity models have become indispensable. These tools offer a rational and cost-effective strategy to triage large chemical libraries by excluding candidates with suboptimal safety or pharmacokinetic profiles, thereby streamlining the identification of viable drug-like molecules in early discovery pipelines [36,37,38].

Building on this foundation, the ability to predict a compound’s interaction profile, whether toward therapeutically relevant targets or off-targets associated with adverse effects, represents the cornerstone of modern virtual screening strategies [39]. The true sophistication of computational triage lies in its capacity to simulate, at scale, molecular recognition events across diverse biological systems, effectively narrowing vast chemical libraries into a manageable set of high-potential candidates [40,41]. Public databases such as ChEBI host an expanding catalog of structurally diverse molecules, many of which are deposited with intended or suspected bioactivity, including a large repertoire of plant-derived compounds [42]. Among these, monoterpenoids stand out as a prolific and historically relevant class of phytochemicals with broad pharmacological potential. These compounds are deeply embedded in ethnopharmacology and are increasingly the subject of preclinical and clinical investigation due to their pleiotropic biological effects [2]. However, despite their promise, monoterpenoids are also characterized by a well-documented set of pharmacokinetic and toxicological limitations. Their small size and high lipophilicity often result in a promiscuous binding profile, increasing the risk of off-target effects and toxicity. Such challenges underscore the importance of applying rigorous in silico filters in early discovery phases to identify candidates with optimized ADMET properties. Computational modeling thus emerges as a tool for prediction and a gatekeeper for safe and effective phytochemical drug discovery [28].

Within our dataset of 1175 monoterpenoid compounds, log P was the most discriminative parameter during the physicochemical screening phase, serving as the primary exclusion factor. This outcome is consistent with the intrinsic high lipophilicity of monoterpenoids, which poses significant challenges to drug-likeness [43]. Excessively lipophilic compounds typically suffer from poor aqueous solubility, which compromises oral absorption and bioavailability, while simultaneously exhibiting high plasma protein binding, leading to limited free drug concentration and unpredictable pharmacokinetics [44,45]. In contrast, only a small fraction of molecules was excluded based on criteria related to molecular weight, number of nHA, nHD, or TPSA, as these physicochemical descriptors tend to fall within acceptable ranges for small molecules such as monoterpenoids [46]. When evaluating absorption-related parameters, the percentage of compounds excluded by HIA prediction, calculated using the pkCSM platform, was moderate. This suggests that the HIA model implemented in pkCSM may be more permissive or uniquely weighted, particularly given its reliance on graph-based signatures and curated datasets. However, results from ADMETlab 2.0, which includes Caco-2 permeability as a key predictive feature, strongly aligned with the HIA trends observed via pkCSM, reinforcing the reliability of both platforms in absorption modeling. The interaction with P-glycoprotein (P-gp), both in terms of substrate recognition and inhibition potential, also played a crucial role in defining absorption liabilities [47]. Monoterpenoid molecules with elevated log P or TPSA values frequently exhibited predicted interactions with P-gp classes I and II, which are associated with efflux-mediated drug resistance and reduced bioavailability [48]. These findings highlight the interplay between physicochemical extremes, particularly excessive lipophilicity and unfavorable absorption characteristics, reinforcing the value of integrated ADMET profiling in early-stage compound triage [49].

Given that the majority of monoterpenoids with extreme log P and log D values were excluded in the early stages of filtering, only a small subset of compounds displayed PPB values above 90%. Nonetheless, when analyzing this parameter in isolation, it becomes evident that PPB alone could eliminate approximately 400 compounds, with over 50% of the dataset being excluded if a more stringent cutoff of 80% was applied. High PPB values are known to reduce the unbound fraction of a drug available for target engagement, thereby limiting efficacy, especially for compounds intended to act at peripheral sites [50]. On the other hand, the small molecular size and pronounced lipophilicity of monoterpenoids contribute to their enhanced BBB permeability, as reflected in the positive correlation observed between these parameters in our dataset. This characteristic suggests that many monoterpenoids may be inherently suited for central nervous system (CNS) applications, an otherwise challenging pharmacological niche due to the restrictive nature of the BBB. These features underscore the value of applying rigorous computational filters during early-stage screening to support rational design and therapeutic positioning of candidates with CNS relevance. However, before reaching target tissues, these compounds must navigate the metabolic landscape, particularly the cytochrome P_450_ enzyme system. In our study, approximately 450 compounds were predicted to be either substrates or inhibitors of one or more CYP isoforms [51]. While CYP inhibition raises concerns regarding drug–drug interactions and metabolic liabilities, the classification of a compound as a CYP substrate carries its own risks, including the generation of unpredictable secondary metabolites that may possess reduced efficacy, altered pharmacodynamics, or increased toxicity [52]. These metabolic transformations can significantly deviate the compound’s activity from its originally intended target profile, thus reinforcing the critical role of early metabolic liability assessments in virtual screening pipelines.

The toxicological profiling of the dataset was conducted using multiple prediction platforms, which is reflected in the redundancy of columns within the supplementary spreadsheet. Among the most critical filters applied were mutagenicity and carcinogenicity, assessed through both categorical (Yes/No) and probabilistic (0 to 1) models. These criteria, even when analyzed independently, excluded approximately 500 compounds. When combined with the carcinogenicity filter from ADMETlab 2.0, the number of acceptable candidates was reduced to just 262 out of the original 1175 molecules. These toxicological endpoints are considered non-negotiable in early-phase drug development, and in cases of discordant predictions across platforms, a conservative exclusion approach was adopted, prioritizing safety by eliminating any molecule flagged as potentially hazardous by any predictive model. The same rigor was extended to additional toxicity-related parameters, including hepatotoxicity, acute and chronic oral toxicity in rodents, MTD, and predicted interactions with hERG I and II potassium channels, which are key indicators of cardiotoxicity and the risk of QT prolongation-induced arrhythmias [53]. Furthermore, respiratory toxicity and potential interference with renal OCT_2_ transporters were also considered. When the dataset was evaluated solely based on this expanded panel of toxicological filters and applying strict inclusion criteria, only 78 molecules remained from the initial 1175, underscoring the stringency and discriminative power of in silico toxicity modeling. Beyond toxicological assessments, medicinal chemistry filters were applied to further refine the dataset. Well-established guidelines such as Lipinski’s Rule of Five, along with filters from Pfizer, GSK, and the Golden Triangle rule, were integrated to assess compliance with drug-likeness heuristics. A total of 248 compounds were approved under these medicinal chemistry criteria. However, compliance with these rules alone does not guarantee favorable pharmacological potential. Therefore, the QED was employed as an integrative scoring metric. All seven final compounds that passed every previous pharmacokinetic and toxicological filter also presented QED scores above 0.65, with 1.0 being the theoretical maximum, indicating a strong alignment with desirable drug-likeness profiles and further validating their prioritization as lead candidates.

Consistent with our predictive framework, the exclusion of several monoterpenoids illustrates the alignment between in silico toxicological filters and well-documented clinical evidence. Camphor (CID: 2537) was flagged by multiple tools for respiratory and neurotoxicity and violated the Golden Triangle threshold, consistent with the cases of lethal respiratory failure and CNS toxicity, especially in children [3,6,7]. Likewise, citral (CID: 638011) was deprioritized based on predicted genotoxicity and hepatotoxicity [54]. The rejection of 1,8-cineole (CID: 2758) for respiratory toxicity echoes clinical observations of bronchospasm following therapeutic use [55]. Collectively, these cases show that coupling predictive toxicology with empirical knowledge can sharpen compound triage and keep development focused on monoterpenoids with genuine, clinically realistic potential.

Out of the 1175 monoterpenoid compounds evaluated, only seven met all criteria across physicochemical, pharmacokinetic, toxicological, and medicinal chemistry domains. Notably, these top-ranked candidates—15-hydroxyculmorin (CID: 98642738), 12-hydroxyculmorin (CID: 139587999), Xylariterpenoid I (CID: 156582074), Xylariterpenoid H (CID: 139205363), Menthoxypropanediol (CID: 5362595), Trichobasabolin F (CID: 146683370), and Vulgarole (CID: 101416067)—do not converge around a single molecular framework. Instead, they display significant structural heterogeneity, indicating that within this class, no single architecture predominates in terms of drug-likeness. This structural variety is reflected in the diversity of ring systems. The final set includes rigid polycyclic scaffolds (e.g., 15-hydroxyculmorin and 12-hydroxyculmorin), flexible monocyclic or open-chain compounds (e.g., Menthoxypropanediol and Trichobasabolin F), and distinct fused-ring structures (e.g., Vulgarole). SAR mapping identified shared core fragments between some candidates, such as the culmorin derivatives, yet no universal scaffold was associated with optimal ADMET profiles, reinforcing the versatility of monoterpenoid space. All molecules were retrieved in their canonical forms from ChEBI and PubChem, and only those passing the full in silico triage were further analyzed for stereoisomeric diversity. Each compound was explored in its possible cis/trans and R/S configurations, and the most stable or bioactive conformer was retained. While stereochemistry was not part of the initial filtering strategy, this final refinement step revealed that small configurational shifts may impact pharmacodynamic interactions, especially in chiral binding environments.

From a physicochemical standpoint, all seven compounds displayed molecular weights between 212 and 254 Da, TPSA values ranging from 37.3 to 49.7 Å^2^, and log P values between 1.7 and 2.9, consistent with favorable passive permeability and oral absorption [56,57]. Importantly, they exhibited low predicted toxicity across all evaluated endpoints, including mutagenicity, carcinogenicity, hepatotoxicity, and cardiotoxicity (via hERG I/II inhibition models). None were flagged by any PAINS or chelating filters, and all complied with Lipinski, Pfizer, GSK, and Golden Triangle medicinal chemistry rules. Drug-likeness was further confirmed through QED scores, with all molecules exceeding the threshold of 0.65, indicating alignment with clinical development profiles [58].

Among the seven top-ranked candidates prioritized in this study, the majority remain virtually unexplored in the biomedical literature. Vulgarole has been reported as a natural constituent in certain aromatic plant species, yet no validated pharmacological or mechanistic data are currently available [59]. The xylariterpenoids and trichobisabolin F, isolated from endophytic and soil-dwelling fungi, such as Xylaria and Trichoderma, are catalogued primarily through phytochemical surveys, with little to no insight into their bioactivity, metabolic stability, or toxicological profiles [11,60,61]. The remaining compounds, including two highly saturated tricyclic diols (CID 98642738 and 139587999) and an alkyl ether-bearing monoterpenoid (CID 5362595), are structurally defined in public repositories but have poor, if any, experimental or pharmacological annotation to date [62,63]. While several of these candidates exceed the canonical ten-carbon definition of monoterpenes, their biosynthetic origin and core architecture remain consistent with monoterpenoid frameworks.

Given the limited characterization of these specific monoterpenoid scaffolds in the current literature, we conducted target prediction analyses to identify putative biological targets for the seven top-ranked compounds. Such an approach is well-established in the context of CADD and is particularly valuable for chemically diverse and underexplored molecular classes [64,65,66,67]. Despite the marked structural heterogeneity among the selected compounds, several high-confidence targets (prediction probability >70%) were shared, suggesting convergence toward common biological pathways. Notably, iNOS and APOBEC-3A emerged as recurrent high-affinity targets across all molecules. Muscarinic acetylcholine receptors (M1, M2, and M4) were also frequently predicted, pointing to a possible modulatory role within the cholinergic system, which could vary depending on tissue-specific receptor expression. Additionally, acetylcholinesterase and the cannabinoid receptor type 2 showed consistent interaction patterns, raising the hypothesis of broader neuroimmune or neuromodulatory activity. Although several terpenoids have been reported to interact with muscarinic and cannabinoid receptors [68,69], functioning as agonists, antagonists, or partial modulators, the specific modes of action for the molecules identified in this study remain unknown. It is not yet clear whether these compounds act via orthosteric agonism, allosteric modulation, enzyme inhibition, or receptor antagonism. The goal of this work was to identify promising molecular targets as a starting point for more detailed mechanistic studies. Given the variety of predicted targets and potential polypharmacological effects, their associated therapeutic or adverse consequences must be considered. Addressing these questions will require a multi-tiered investigative strategy, beginning with in silico tools, such as docking and molecular dynamics [70], and progressing through rigorous in vitro and in vivo validation [24,25]. Such advances, however, are only possible through comprehensive screening studies like the present work, which systematically mine chemical space, triage promising scaffolds, and provide a rational foundation for future experimental investigation [71].

Collectively, these seven structures represent a previously overlooked chemical space within this class, distinct in both scaffold diversity and predicted ADMET favorability. Their emergence from a rigorous, multilayered in silico triage underscores their compliance with key medicinal chemistry thresholds and their untapped potential as scaffolds for future drug development efforts. As such, these molecules constitute promising leads for experimental validation and rational derivatization in therapeutic contexts.

## 4. Materials and Methods

### 4.1. Compound Retrieval and Dataset Preparation

The SMILES (Simplified Molecular Input Line Entry System) representations of all molecules classified as monoterpenes were retrieved from the Chemistry Database Managed by the National Institutes of Health—PubChem (https://pubchem.ncbi.nlm.nih.gov/, accessed on 5 May 2025). The selection process utilized the Chemical Entities of Biological Interest (ChEBI) classification [42], specifically navigating through the ChEBI Ontology to the class “Monoterpenoids”. All identified monoterpenoid compounds were compiled and archived prior to the in silico analyses.

### 4.2. Computational Tools and Platforms

The compiled molecules were analyzed using a suite of computational platforms for QSAR modeling, ADMET predictions, medicinal chemistry profiling, and toxicity and physicochemical characterization (Appendix A). In total, 12 computational tools were employed, including pKCSM (https://biosig.lab.uq.edu.au/pkcsm/prediction, accessed on 5 February 2025) [72], ADMETlab 2 (https://admetmesh.scbdd.com/, accessed on 5 February 2025) [73], SwissADME (http://www.swissadme.ch/, accessed on 6 February 2025) [74], Molinspiration Cheminformatics (https://www.molinspiration.com/, accessed on 12 February 2025) [75], AI Drug Lab (https://ai-druglab.smu.edu/, accessed on 12 February 2025) [76], CODD-PRED (https://fca_icdb.mpu.edu.mo/codd/, accessed on 12 February 2025) [77], PRED-hERGG (https://predherg.labmol.com.br/, accessed on 20 February 20) [78], XenoSite (https://xenosite.org/, accessed on 20 February 2025) [79], VNN-ADMET (https://vnnadmet.bhsai.org/vnnadmet/login.xhtml, accessed on 20 February 2025) [80], ToxinPRED (https://webs.iiitd.edu.in/raghava/toxinpred/algo.php, accessed on 20 February 2025) [81], ADMETopt (https://lmmd.ecust.edu.cn/admetsar2/admetopt/, accessed on 20 February 2025) [82], and ProTox-III (https://tox.charite.de/protox3/, accessed on 5 May 2025) [35].

### 4.3. Pharmacokinetic Profiling

The physicochemical properties were assessed according to established thresholds predictive of drug-likeness and bioavailability. Molecular weight (MW) was constrained between 100 and 600 Daltons, as compounds within this range generally exhibit favorable membrane permeability and oral bioavailability [83]. Molecular volume was considered optimal between 100 and 300 Å^3^, balancing permeability and solubility [84]. Density values typically range from 0.8 to 1.5 g/cm^3^ for small-molecule drugs, reflecting appropriate molecular compactness [85]. Molecular flexibility was assessed by the number of rotatable bonds, with compounds having ≤10 rotatable bonds considered preferable, given that excessive flexibility can impair oral bioavailability and metabolic stability [86]. The number of hydrogen bond acceptors (nHA) and donors (nHD) was evaluated, following the conventional Lipinski’s Rule of Five, which recommends no more than 10 acceptors and 5 donors to optimize membrane permeability [87]. The topological polar surface area (TPSA) was parameterized, with an upper limit of 140 Å^2^, above which passive diffusion across biological membranes is significantly hindered [88]. The lipophilicity of compounds was estimated using LogP, with ideal values falling between −0.4 and +5.6 and a preferred range of 1 to 3 to ensure balanced hydrophilic and lipophilic characteristics [89]. Additionally, LogD at physiological pH (7.4) was considered, particularly for ionizable compounds, to predict their solubility and permeability profiles under biological conditions [90].

For absorption-related parameters, the gastrointestinal absorption (GIA) was predicted, with compounds exhibiting high absorption favored for potential oral administration. The Caco-2 cell permeability assay was used as a surrogate for intestinal permeability, with values greater than 0.9 × 10^−6^ cm/s indicating good passive diffusion potential [91]. Aqueous solubility was also estimated, with compounds showing LogS values above −4 considered adequately soluble for bioavailability purposes, as solubility below −6 is typically indicative of poor solubility [92]. Skin permeability was analyzed via Log Kp, where values between −3 and −6 cm/s represent the expected range for systemic absorption [93]. Furthermore, compounds were evaluated for their status as substrates or inhibitors of P-glycoprotein (P-gp), particularly types I and II, as interaction with this efflux transporter can significantly impact oral bioavailability and drug resistance profiles; ideally, compounds should not be P-gp substrates or inhibitors to minimize these risks [94].

Plasma protein binding (PPB) was evaluated, with compounds showing binding levels below 90% preferred to ensure an adequate free fraction (Fu) capable of exerting pharmacological activity; high PPB (>95%) can substantially reduce the free, active drug concentration [95]. The volume of distribution at steady state (VDss) was considered, with values typically ranging from 0.1 to 1 L/kg, indicating favorable tissue distribution without risk of bioaccumulation [96]. Additionally, blood–brain barrier (BBB) permeability was assessed via logBB, with values above −1 indicative of sufficient CNS penetration, while central nervous system (CNS) permeability was estimated using logPS, where values greater than −3 suggest a likelihood of effective CNS exposure.

In the context of metabolism, compounds were evaluated for their potential to inhibit or act as substrates of major cytochrome P_450_ (CYP) isoforms, namely CYP_1A2_, CYP_2C19_, CYP_2C9_, CYP_2D6_, and CYP_3A4_ [97]. Ideally, compounds should neither inhibit nor be substrates for these enzymes to minimize the risk of drug–drug interactions and variable metabolic clearance [98]. The excretion potential was analyzed by predicting total systemic clearance, with values in the range of 1 to 10 mL/min/kg generally considered acceptable to balance efficacy and safety. Additionally, the potential for interaction with the renal organic cation transporter 2 (OCT_2_) was evaluated, as compounds that do not interact with OCT_2_ are less likely to cause nephrotoxic effects or alter renal drug elimination [99].

### 4.4. Toxicological Risk Assessment, Drug-likeness, and Medicinal Chemistry Filters

The mutagenic potential was computationally predicted using an Ames test model, where a negative result is considered favorable [100]. Carcinogenicity and hepatotoxicity were also predicted, with non-carcinogenic and non-hepatotoxic profiles representing favorable safety attributes. Skin sensitization potential was examined, with negative predictions indicating reduced risk for topical formulations. Acute and chronic toxicity in rodent models was assessed, along with the maximum tolerated dose (MTD) in humans, to inform the potential safety margins. Cardiac safety was evaluated by predicting hERG channel inhibition (types I and II), where non-inhibition is essential to minimize the risk of arrhythmogenic cardiotoxicity [78]. Finally, respiratory toxicity was predicted, with negative outcomes preferred to ensure respiratory safety. In addition to these pharmacokinetic and toxicological evaluations, all compounds were assessed for drug-likeness using established rules and filters, including Lipinski’s Rule of Five (≤1 violation), Pfizer’s and GlaxoSmithKline (GSK) filters, the Golden Triangle criterion (balancing LogP and MW for clinical success), and Veber’s rule (TPSA ≤ 140 Å^2^ and rotatable bonds ≤ 10) to evaluate their overall potential as viable drug candidates [101,102,103].

### 4.5. Data Integration and Filtering Strategy

Following the retrieval of all data from the aforementioned computational platforms, a comprehensive Excel spreadsheet was constructed, encompassing a total of 54 (Appendix A) distinct parameters analyzed for each monoterpenoid compound obtained from PubChem. Systematic inclusion and exclusion criteria were applied to this dataset, wherein only those molecules that met the predefined thresholds for each specific parameter, based on physiologically relevant and pharmacologically acceptable ranges, were retained. This stepwise filtering strategy ensured that compounds progressing through each phase conformed to established drug-likeness and safety profiles. The application of these parameter-specific filters served as the primary method for screening and prioritizing the most promising molecular candidates. This rigorous approach was systematically extended across all 54 parameters, ensuring that the final subset of molecules exhibited optimal profiles across physicochemical, pharmacokinetic, and toxicological domains, thereby identifying the most suitable candidates for potential drug development.

Following this stage, the entire dataset, comprising all computed descriptors, was imported into the software DataWarrior v5.5.0 [104] for statistical analysis and bioprospecting. The Bravais–Pearson correlation coefficient (R) was also calculated to assess the strength and direction of the linear relationship between the analyzed variables. Additionally, an automatic structure–activity relationship (SAR) analysis was conducted using DataWarrior’s integrated tools, aimed at identifying structural features associated with bioactivity predictions.

### 4.6. Target Bioactivity Prediction

To predict the putative molecular targets of the most promising compounds, the TargetNet platform was employed [105]. The SMILES representations of the selected molecules were submitted using the following configuration: fingerprint type set to ECFP4, and only models with area under the curve (AUC) ≥ 0.7 were included to ensure high-confidence predictions. A probability cutoff threshold of 70–100% was applied to retain only target interactions with strong predicted likelihood, thereby increasing the biological relevance of the output. This analysis enabled the identification of potential protein targets for each compound, offering mechanistic insights and guiding hypothesis-driven validation in future studies.

## 5. Conclusions

This study applied a rigorous, multi-parameter in silico strategy to evaluate 1175 monoterpenoid compounds based on physicochemical, pharmacokinetic, toxicological, and medicinal chemistry criteria. Seven structurally diverse candidates emerged from successive computational filtering steps, each exhibiting favorable drug-likeness profiles, QED scores above 0.65, and no predicted toxicological liabilities. Target prediction analyses revealed a broad range of potential biological interactions, including enzymes, GPCRs, nuclear receptors, and oxidoreductases, indicating polypharmacological potential within this underexplored chemical space. While these findings highlight promising properties, the lack of experimental characterization underscores the need for validation. As such, in vitro studies are warranted as a first step to confirm bioactivity, refine target engagement hypotheses, and support further progression in the drug discovery pipeline.

## Figures and Tables

**Figure 1 ijms-26-07671-f001:**
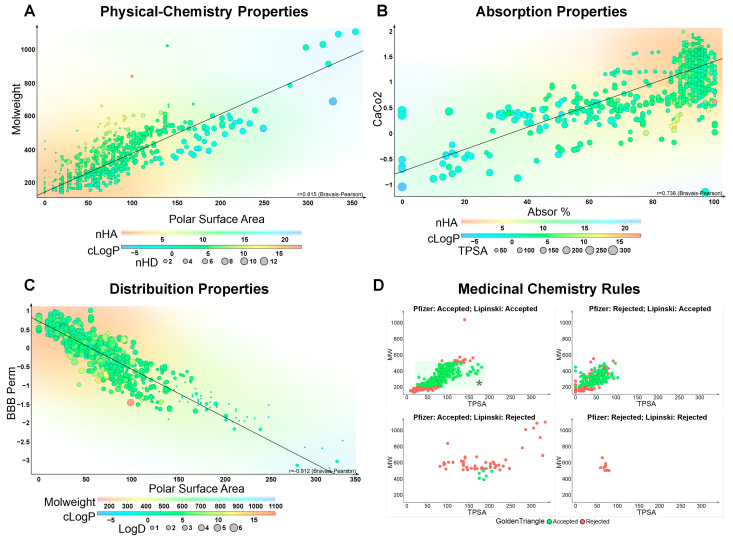
Physicochemical, absorption, distribution, and medicinal chemistry properties of screened monoterpenoids. (**A**) Correlation (r = 0.815, Bravais–Pearson) between molecular weight and topological polar surface area (TPSA), colored by hydrogen bond acceptors (nHA) and donors (nHD), with point size representing calculated LogP (cLogP). (**B**) Correlation between Caco-2 cell permeability and intestinal absorption (%), with markers colored by nHA and cLogP and sized by TPSA. A moderate positive correlation is noted (r = 0.736, Bravais–Pearson). (**C**) Relationship between blood–brain barrier permeability (Log BB) and TPSA, with inverse correlation (r = −0.912, Bravais–Pearson), supporting the predictive role of TPSA in CNS penetration. (**D**) Classification of molecules according to medicinal chemistry filters: compliance with Lipinski and Pfizer rules and positioning within the Golden Triangle region. Green points indicate accepted molecules; red points were excluded by respective criteria. The best compounds are highlighted (*). The analysis was performed using DataWarrior software.

**Figure 2 ijms-26-07671-f002:**
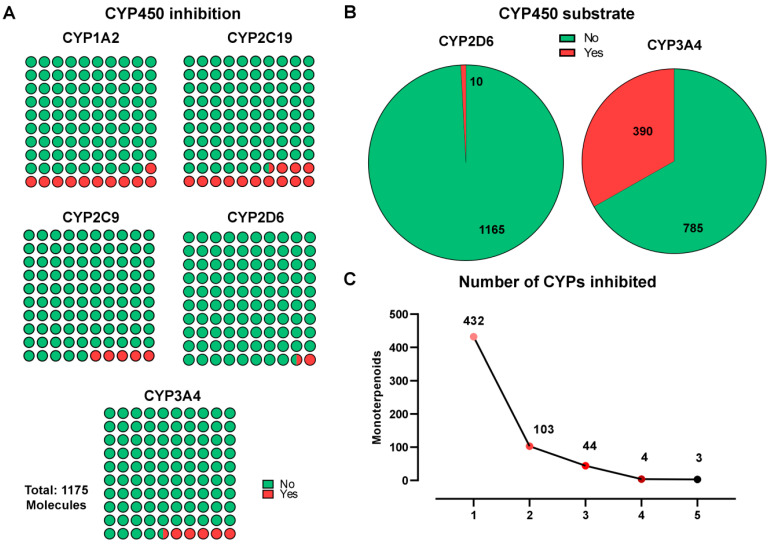
Predicted interactions of monoterpenoid compounds with cytochrome P450 (CYP) enzymes. (**A**) Classification of all 1175 monoterpenoids according to their predicted inhibitory activity compared with five major CYP isoforms: CYP_1A2_, CYP_2C19_, CYP_2C9_, CYP_2D6_, and CYP_3A4_. Green indicates no inhibition; red indicates predicted inhibition. (**B**) Proportions of monoterpenoids predicted to act as substrates of CYP_2D6_ and CYP_3A4_, represented by pie charts. Most compounds were predicted not to be substrates, with CYP_3A4_ having the highest substrate rate (33.2%). (**C**) Distribution of the number of CYP isoforms inhibited per compound. The majority of monoterpenoids inhibited one CYP, while only a small subset inhibited four or five simultaneously. The analysis was performed using GraphPad Prism 8 software.

**Figure 3 ijms-26-07671-f003:**
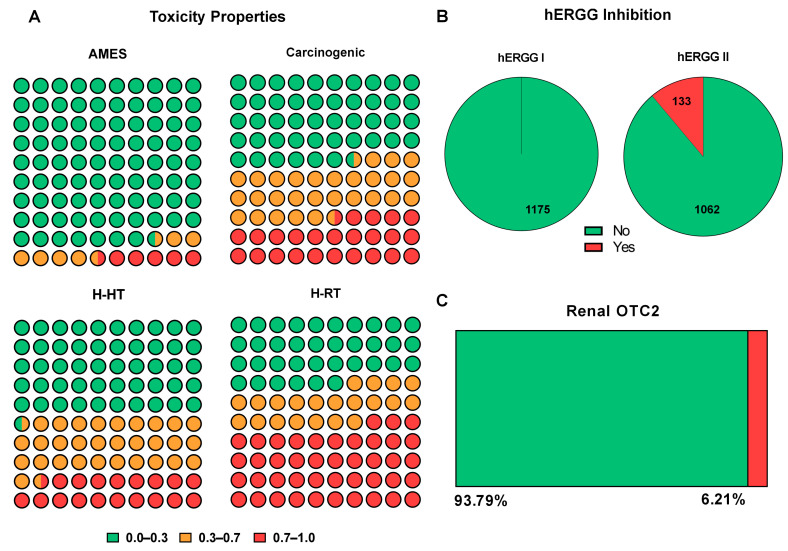
Toxicological predictions for monoterpenoid compounds. (**A**) Classification of 1175 molecules based on predicted toxicological endpoints. AMES test (mutagenicity), carcinogenicity, human hepatotoxicity (H-HT), and human respiratory toxicity (H-RT) are represented by color-coded probability ranges: green (0.0–0.3), yellow (0.3–0.7), and red (0.7–1.0). (**B**) Predicted inhibition of cardiac hERG potassium channels. None of the compounds were predicted to inhibit hERG I, while 133 molecules showed potential inhibition of hERG II. (**C**) Predicted interaction with the renal organic cation transporter 2 (OCT_2_). The analysis was performed using GraphPad Prism 8 software.

**Figure 4 ijms-26-07671-f004:**
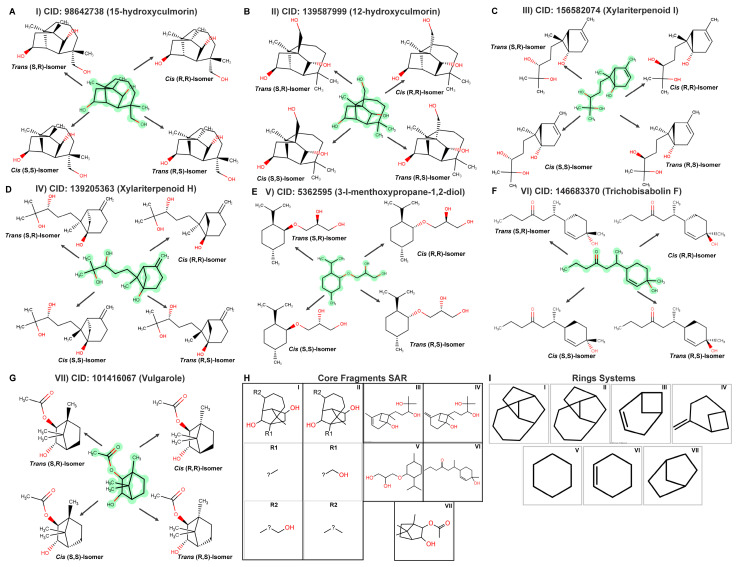
Stereochemical and structural analysis of the seven top-ranked monoterpenoid compounds. (**A**–**G**) Stereoisomeric configurations (cis/trans and R/S) of each selected compound, generated using MarvinSketch (ChemAxon); The most stable or bioactive conformer is shown in green as a semi-transparent 3D surface overlay on the Ball and Stick model, illustrating its spatial alignment with the reference molecular structure. Compounds are identified by their PubChem CIDs and common names: (**A**) I: 98642738 (15-hydroxyculmorin), (**B**) II: 139587999 (12-hydroxyculmorin), (**C**) III: 156582074 (Xylariterpenoid I), (**D**) IV: 139205363 (Xylariterpenoid H), (**E**) V: 5362595 (Menthoxypropanediol), (**F**) VI: 146683370 (Trichobasabolin F), and (**G**) VII: 101416067 (Vulgarole). (**H**) Core fragments extracted using DataWarrior for structure–activity relationship (SAR) analysis, highlighting conserved scaffolds and variations at substituent positions R1 and R2. (**I**) Representative core ring systems identified among the prioritized compounds, classified into tricyclic (I and II, identical), bicyclic (III and IV, identical; VII, distinct), and monocyclic (V and VI, identical) frameworks. This highlights the structural diversity spanning fused, bridged, and unsaturated carbocyclic motifs. The "?" symbol indicates the core structural fragment of the molecule.

**Figure 5 ijms-26-07671-f005:**
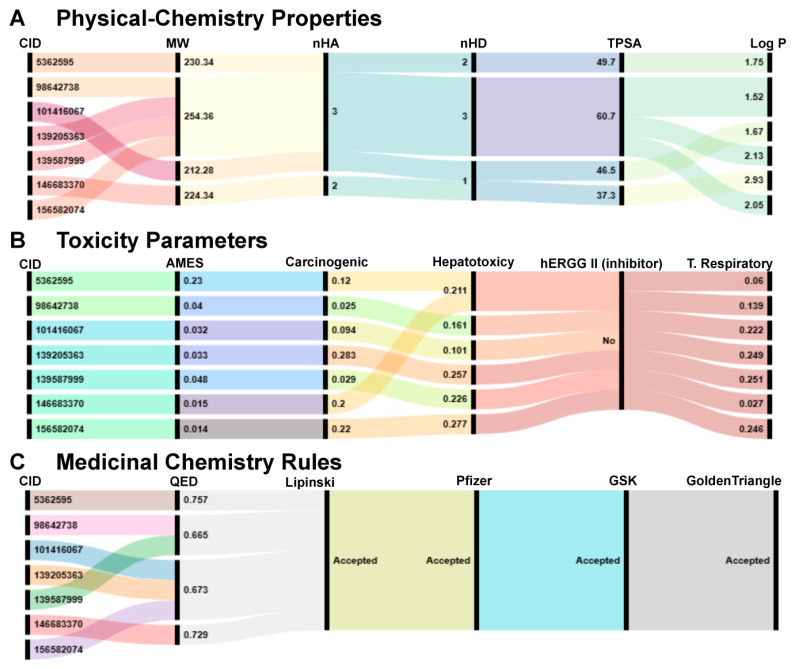
In silico evaluation of physicochemical, toxicity, and medicinal chemistry properties of the seven top-ranked candidate compounds. (**A**) Physicochemical properties, including molecular weight (MW), number of hydrogen bond acceptors (nHA), number of hydrogen bond donors (nHD), topological polar surface area (TPSA), and calculated lipophilicity (Log P), are shown for each compound identified by its PubChem CID. (**B**) Toxicological profile comprising results from AMES mutagenicity prediction, carcinogenicity potential, hepatotoxicity, inhibition of the hERG II potassium channel, and respiratory toxicity. (**C**) Compliance with medicinal chemistry rules, including quantitative estimate of drug-likeness (QED) and filters from major pharmaceutical companies (Lipinski (Washington, DC, USA), Pfizer (New York, NY, USA), GSK (Durham, NC, USA), and the Golden Triangle rule (Washington, DC, USA)).

**Figure 6 ijms-26-07671-f006:**
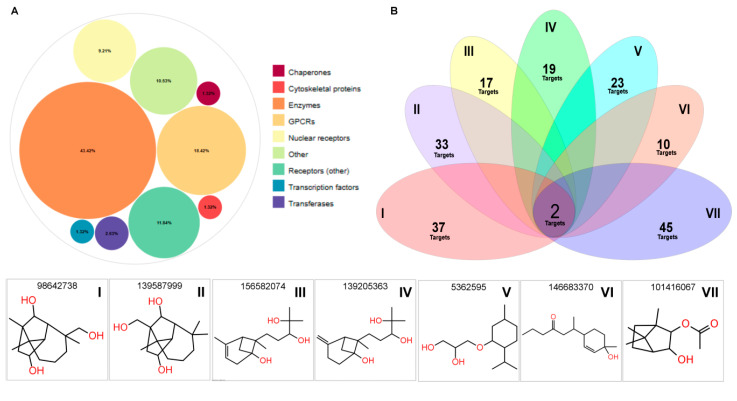
Classification of molecular targets and shared target analysis of the seven top-ranked monoterpenoid compounds. (**A**) Bubble plot representing the distribution of predicted molecular targets according to their functional classes. The majority of targets are enzymes (43.42%), followed by GPCRs (18.42%). Minor classes include nuclear receptors, transcription factors, transferases, cytoskeletal proteins, chaperones, and others. (**B**) Venn diagram showing the number of unique and overlapping predicted targets among the seven compounds (I–VII), highlighting two common targets shared across all molecules. Bottom panel: Two-dimensional chemical structures of the seven final compounds, identified by their PubChem CIDs and labeled I–VII for cross-reference.

**Figure 7 ijms-26-07671-f007:**
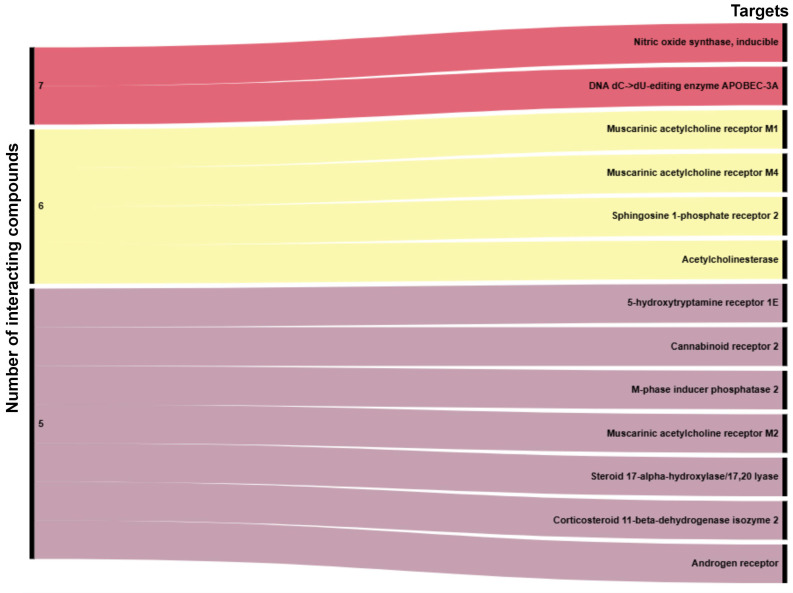
Sankey diagram illustrating the number of compounds predicted to interact with each molecular target. Nitric oxide synthase, inducible (iNOS), and DNA dC→dU-editing enzyme APOBEC-3A were predicted as common targets for all seven compounds. Targets interacting with six molecules include muscarinic acetylcholine receptors M1 and M4, sphingosine 1-phosphate receptor 2, and acetylcholinesterase. A group of targets was identified as shared by five molecules, including 5-hydroxytryptamine receptor 1E, cannabinoid receptor 2, M-phase inducer phosphatase 2, muscarinic acetylcholine receptor M2, and multiple steroid-related enzymes. These targets represent potential key nodes in the pharmacological network of the selected monoterpenoids.

## Data Availability

The original contributions presented in this study are included in the article/Appendix A. Further inquiries can be directed to the corresponding author(s).

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
