# Peer review of "Computational Profiling of Monoterpenoid Phytochemicals: Insights for Medicinal Chemistry and Drug Design Strategies"

_ijms, 2025, doi:10.3390/ijms26167671_

Round 1
Reviewer 1 Report
Comments and Suggestions for Authors
1.- typographical mistakes should be reviewed.
2.- Please highlight the novelty of the study. In fact, most of the predictions described are usually reported in computational studies.
3.- The “SAR” analysis is too weak. I suggest that SAR studies focus on how the chemical structure impacts the binding to the two targets through docking studies. Please, do not include the description of the chemical structures of 7 ligands as SAR.
4.- Docking studies are not provided for the best 7 ligands against the two targets. This could strengthen the study.
5.- The discussion is poorly described. Are the correlations between physicochemical properties and ADMET predictions clear? It also addresses iNOS as a potential drug target and its implications. Additionally, there is a lack of comparison to other successful studies on the relationship between property prediction (physicochemical, ADMET) and biological response (bioactivity).
6.- Authors mentioned that LogP has the most discriminative factor; thus, authors should explain why this parameter was not used as the first filter.
7.- Nowadays, the Lipinski rule is not decisive in excluding molecules as promising candidates. Please update.
https://doi.org/10.1016/j.addr.2016.04.029
https://doi.org/10.1021/acs.jmedchem.8b00686
8.- Authors mention that ligands remain unexplored; however, this is not entirely true.
https://doi.org/10.1002/ps.6160
https://doi.org/10.1016/s0378-5173(03)00205-9
9.- The thirteenth computing platform is missing. Only 12 have been cited.
10.- fizer and GlaxoSmithKline filters, and the Golden Triangle criterion, should be appropriately cited by providing access online.
11.- Conclusions should be rewritten. Preclinical applications are too premature; please suggest in vitro validation as a first step.
8.- My final comment is that the study design is not novel; it would have been more informative to compare the ADMET properties of 1,175 terpenes with a multi-parameter computational workflow and determine if the results lead to the same seven best terpenes. It would emphasize the importance, robustness, and novelty of the multi-parameter computational study described by the authors.
Author Response
We sincerely thank the reviewer for their time, careful assessment, and constructive feedback, which have contributed meaningfully to the improvement of our manuscript. All comments that required a point-by-point response have been thoroughly addressed below. Where appropriate, we have incorporated the recommended changes directly into the revised manuscript. In other cases, we provide detailed justifications when changes were not implemented, ensuring transparency and scientific reasoning. We trust that these revisions have enhanced the overall clarity, robustness, and quality of the work.

Reviewer 2 Report
Comments and Suggestions for Authors
The authors present the manuscript titled " Computational Profiling of Monoterpenoid Phytochemicals: Insights for Medicinal Chemistry and Drug Design Strategies, " where they select 1175 monoterpenoids from a public database and apply a rational filter based on their ADMET properties, ultimately predicting potential molecular targets. While this initial approach is sound, the methodology section mentions 12 free QSAR tools used to predict different parameters for screening. However, the results and discussion sections do not specify which software was used for each pharmacological attribute, making reproducibility difficult. Furthermore, with only seven final candidates, less than 1% of the total, that exhibit desirable properties, a structure-based drug design SBDD approach could have been employed to examine their interactions with specific targets and provide deeper insight into their potential. This manuscript represents only half of a complete in silico development and, in its current form, should not be published in this form.
Additional comments
The introduction should add information regarding CADD
1 The results mentioned between lines 102 and 109 require bibliographic references.
2 Figure 2 has very low resolution and needs improvement. The software used for each calculation should be described in the legend.
3 Figure 4 should be redesigned. It is poorly made and neglects key details. It needs to include the stereochemistry of the molecules. If they are racemic, predict the properties of the enantiomers and diastereoisomers separately.
4 Standardize the numbering in section 2.7 versus the table.
5 The discussion should focus on the prediction software and compare them, rather than on how the values fall outside the ranges. This is because the manuscript does not explain why one software is chosen over another or include a comparison between them, especially considering that the entire manuscript is in silico.
Author Response

(The authors gave the same response as above.)

Reviewer 3 Report
Comments and Suggestions for Authors
The comments and suggestions are given in the attached Review.

Author Response

(The authors gave the same response as above.)

Round 2
Reviewer 1 Report
Comments and Suggestions for Authors
Thank you for addressing most of the comments. Just so you know, for future reference, please do not include SAR analysis in in silico studies, as in vitro data are missing. In that case, you may consist of a Structure-Binding Relationship (SBR) analysis.
Thank you!
Reviewer 2 Report
Comments and Suggestions for Authors
The authors made some of thes suggested corrections and argued why not the other corrections. I continuo to think that. SBDD is neccesary.